# Understanding dimensions of trust in AI through quantitative cognition: Implications for human-AI collaboration

Weizheng Jiang[1,2]*, Dongqin Li[2], Chun Liu[2]

1 School of Management, Wuhan University of Science and Technology, Wuhan, China, 2 School of Management, Wuhan Technology and Business University, Wuhan, China

* WeizhengJiang@wust.edu.cn

## Abstract

Human-AI collaborative innovation relies on effective and clearly defined role allocation, yet empirical research in this area remains limited. To address this gap, we construct a cognitive taxonomy trust in AI framework to describe and explain its interactive mechanisms in human-AI collaboration, specifically its complementary and inhibitive effects. Specifically, we examine the alignment between trust in AI and different cognitive levels, identifying key drivers that facilitate both lower-order and higher-order cognition through AI. Furthermore, by analyzing the interactive effects of multidimensional trust in AI, we explore its complementary and inhibitive influences. We collected data from finance and business administration interns using surveys and the After-Action Review method and analyzed them using the gradient descent algorithm. The findings reveal a dual effect of trust in AI on cognition: while functional and emotional trust enhance higher-order cognition, the transparency dimension of cognitive trust inhibits cognitive processes. These insights provide a theoretical foundation for understanding trust in AI in human-AI collaboration and offer practical guidance for university-industry partnerships and knowledge innovation.

## Introduction

AI technology has evolved from being merely a tool to becoming an integral component of human-AI coexistence [1]. "Friendly" human-AI collaboration (HAC) is increasingly emerging as a core driving force of innovation [2]. Related technologies, such as decision-support systems powered by machine learning models and generative AI (GAI) based on large language models, have gained significant attention across industries. The reason lies in their ability to enhance the integration of humans and AI in collaborative processes, thereby shifting task characteristics toward greater non-standardization, non-procedurality, and complexity [3,4]. This trend has also revitalized both the theory and practice of knowledge innovation management,

**Data availability statement:** All relevant data are within the manuscript and its Supporting Information files.

**Funding:** The author(s) received no specific funding for this work.

**Competing interests:** The authors have declared that no competing interests exist.

particularly in how AI assistance significantly enhances human analytical capabilities and creativity.

From a knowledge management perspective, improved collaboration capabilities facilitate cross-disciplinary communication, social network construction, knowledge spillover, and innovation model refinement. As a critical component of organizational competence, collaboration capability is closely linked to organizational adaptability, efficient resource allocation, and technology-driven progress. With the rise of digital transformation and AI-powered knowledge innovation, AI's role in collaboration is gradually evolving into a human-like partnership. One of the key advantages of AI-driven knowledge innovation is that HAC fosters complementary strengths between humans and AI. A notable example is that AI excels at processing large volumes of data, making it more adept at generating objective results, whereas humans possess stronger empathy and a deeper understanding of emotions and tacit knowledge. Compared to AI, this makes humans more skilled in shaping intuition and creativity. Studies have shown that effective HAC is contingent on the level of trust humans place in technology. For instance, Bedué and Fritzsche [5] argued that the adoption of AI technology is closely tied to trust, and that trust in AI directly constrains firms' competitive and operational advantages. Similar conclusions have been drawn by Montag et al. [6].

Trust in AI (TAI) can enhance human willingness to engage with new technologies and better facilitate the advantages of HAC. However, a critical barrier to deep human-AI integration remains: the mechanisms underlying human trust in AI are not yet fully understood (see S1 Table). Although extensive research has demonstrated that TAI directly influences technology adoption, collaboration efficiency, and ethical perceptions [7,8], existing literature largely limits its discussion to "instrumental" trust—such as reliability and transparency—while neglecting the multidimensional trust interactions required for AI as a "collaborative partner" (encompassing emotional and human-like trust) [4]. This fragmentation results in a significant paradox: the more human-like AI becomes, the greater the uncertainty surrounding trust-building mechanisms, which, rather than enhancing HAC, ultimately hinders it [9,10], thereby obstructing innovation. A review of literature since 2023 reveals that research on human-AI coexistence is gaining momentum. Zirar et al. [1] argue that AI and humans exist in a state of coexistence—one that involves complementary strengths, mutual reinforcement, professional skill enhancement, and the establishment of psychological trust. However, the realization of this human-AI coexistence hinges on a crucial prerequisite: the roles and task distribution in HAC must be clearly defined and highly efficient.

Although research on TAI is progressing toward AI as a collaborative partner, a critical issue remains insufficiently addressed: how to better understand the role allocation between humans and AI in decision-making and innovation processes. Existing studies primarily focus on TAI itself, leading to a significant gap in exploring AI's concrete impact on decision-making. For instance, Choung et al. [7]directly examine the multidimensionality of trust and its relationship with technology acceptance in human-AI interaction but fail to clarify how different trust dimensions

support various types of tasks in collaboration. Similarly, studies on related topics [5,11] tend to link trust levels to AI's role recognition in cooperation but do not deeply investigate task allocation and workload distribution in the collaboration process. Moreover, current TAI's dimensions constructs overly rely on technology acceptance models such as TAM (Technology Acceptance Model) [e.g., 5] or derivative models like TRiSM (Trust, Risk, Security Management) [e.g., 12], and HCTAM (Human-Computer Trust Model) [e.g., 13], while lacking sufficient analysis from psychological and organizational behavior perspectives, particularly regarding individual differences. Although TAM and its derivatives are widely applicable, they primarily address AI itself rather than HAC in knowledge innovation. In reality, major decision-making and complex tasks still heavily rely on human expertise. A strong piece of evidence supporting this is the critical role of soft skills in AI-enabled transformations, as demonstrated in the empirical study by Fletcher and Thornton [14]. Finally, general conclusions in HAC research suggest that humans remain irreplaceable[e.g., 1,15,16]. However, these studies have not systematically clarified or categorized the human role in the era of intelligent systems, leaving the fundamental question unanswered: how to define and reinforce humanity's irreplaceable position in innovation.

Therefore, this study integrates the TAM framework with cognitive models to gain insights into the issue of role allocation. Cognition is a crucial representation of human collaboration and innovation activities, serving as the key driving force behind decision-making behavior. By utilizing Bloom's taxonomy of cognitive levels, this research quantifies different cognitive behaviors exhibited by individuals in HAC processes. By comparing the importance of various cognitive levels in HAC, this study identifies differences in AI-enabled cognitive practices among humans. Furthermore, this study empirically examines the complementary advantages of AI and humans while addressing potential limitations, providing a foundation for role allocation in human-AI coexistence. These findings offer more effective guidance for AI-driven knowledge innovation.

This study surveyed 408 university interns from 2022 to 2023 to assess their level of TAI, thereby analyzing their HAC relationships. By linking TAI with cognitive levels, the study determines role configurations in HAC. These findings provide valuable insights into digital transformation, university-industry collaboration, and human resource management.

## Theoretical foundations

### The impact of AI utilization on cognition

AI has a profound impact on cognitive processes, encompassing both positive and negative effects. AI positively impacts cognitive acquisition in several ways. First, AI addresses a large volume of procedural and repetitive cognitive tasks, thereby creating conditions for the acquisition of complex and breakthrough cognition [17]. Second, the use of AI enhances individuals' ability to decode abstract knowledge, thereby improving the efficiency of knowledge conversion [18]. Moreover, AI utilization not only enhances individual cognitive abilities but also significantly improves team cognition and insights through analysis, evaluation, and creation [19]. Most importantly, in dynamic and complex environments, TAI affects the enhancement of cognitive abilities in HAC [20].

However, there are also negative aspects to the enhancement of cognitive abilities through AI utilization. Carr [21] explicitly stated in his book *What the Internet Is Doing to Our Brains* that information technologies like the Internet overload cognitive load, leading to superficial thinking. This viewpoint includes the negative impact of technologies such as AI on memory and understanding cognition. Moreover, for individuals with limited adaptability to new technologies, low proactivity, and a strong reliance on established paths, the integration of AI into the collaboration process may still fail to enable them to gain meaningful insights [14,22].

Therefore, this study adopts a cognitive perspective to develop a quantifiable cognitive model, aiming to uncover the complementary mechanisms in HAC through the lens of TAI. Additionally, it explores specific aspects in which AI may potentially diminish human cognitive abilities. The ultimate goal is to elucidate the role allocation within HAC.

                                                                 

## Cognitive quantification and human-AI collaboration

Bloom's cognitive taxonomy provides actionable theoretical labels for quantifying the knowledge collaboration and innovation process. This framework is logically consistent with Nonaka's [23] SECI knowledge spiral model: the SECI model emphasizes innovation through the dynamic cycle of "socialization-externalization-combination-internalization," while Bloom's taxonomy provides a hierarchical measurement tool for individual cognitive behaviors in this process. In the "socialization" and "externalization" stages of SECI, individuals rely on memory and understanding to decode tacit knowledge; in the "combination" and "internalization" stages, application, analysis, and creation play a key role in the tacitification of explicit knowledge. This study seeks to reveal a key contradiction: although AI technology can accelerate the externalization of knowledge (such as information retrieval and data visualization), it may inhibit the internalization process of higher-order cognition (such as reflection and critique) [9]. HAC systems enhance collaboration efficiency by externalizing tacit knowledge [24], but their effect on cognitive levels is bidirectional: they may either free human resources to focus on higher-order innovation or lead to "cognitive dependence," weakening deeper thinking abilities [25,26].

Bloom's taxonomy has been widely and successfully applied in the field of education, offering valuable insights for knowledge innovation management. First, by quantifying the proportion of different cognitive levels, it facilitates the construction of individual and team competency models [27]. Second, the development of Bloom's taxonomy and its digital counterpart aims to support the exploration of digital and informal learning pathways [28]. Furthermore, the scaffolding design framework of digital Bloom's taxonomy provides a basis for defining roles in HAC.

However, this evaluation approach also has limitations. First, the application of Bloom's taxonomy has been primarily focused on the field of education, and its dynamism and contextual adaptability in knowledge management scenarios have not been fully validated. Additionally, existing studies on this model do not address TAI and HAC, making it difficult to explain how human-AI relationships regulate and influence cognition. For example, it remains unclear whether anthropomorphic trust enhances AI's support for cognitive behaviors such as understanding and application.

To overcome these limitations, this study proposes the following approaches: integrating Bloom's taxonomy with the SECI model and constructing a cognitive quantification framework for knowledge management using the AAR method; incorporating dimensions of TAI to analyze their dynamic relationships with different cognitive levels; and employing machine learning to capture nonlinear relationships in HAC, thereby surpassing the constraints of traditional linear assumptions.

## Dimensions of trust in AI

**Human-like trust.** Human-like trust (HLT) is established through benevolence, reliability, and the degree of anthropomorphism [8,29]. The formation of this trust is accompanied by human experience accumulation and technological advancements [30]. From the perspective of intelligent system design, Jung et al. [31] pointed out that participants' risk-taking ability and collaborative learning processes significantly influence the establishment of HLT. In their study on sharing economy platforms, Califf et al. [32] found that user familiarity and transparent feedback are key drivers of HLT formation. Additionally, research suggests that the adaptation of AI systems to social and cultural values—such as fairness, embodiment, and empathy—also plays a crucial role in building HLT [7].

**Functionality trust.** Functionality trust (FT) is related to the technology itself and its compatibility with specific scenarios. It is established through factors such as functional practicality, functional integrity [33,34], predictability, and transparency [34], as well as user experience, comprehension, and the characteristics of different task scenarios [4]. Additionally, FT is gradually formed based on user interactions with technology, and individuals can develop this trust through both direct and indirect experiences [35].

**Cognitive trust.** Cognitive trust (CT) in AI is established through reliability, transparency, and accessibility. Unlike HLT and FT, reliability in CT is associated with the consistency of AI performance, which is built by comparing AI's behavior at a given moment and over time [5,11]. In the context of CT, transparency reflects the depth of human reasoning and

understanding in AI-assisted decision-making [36]. Riley and Dixon [37] also suggest that transparency is closely linked to AI dependence. Accessibility, on the other hand, stems from user experience and attitudes toward technology, manifesting as explainability, perceptibility, and intuitiveness [7].

**Emotional trust.** Emotional trust (ET) in AI involves users' perception of AI's goodwill and emotional connection. Based on a review of relevant literature, ET is primarily established through two aspects: social factors (such as social networks, cultural differences, perceptions of social moral norms, fairness, and self-authority) and human-AI emotional interaction (including humans' perception and understanding of AI's goodwill, as well as AI's support for human emotions) [37–39]. In summary, ET highlights both the role of technological advancements in trust formation and the adaptability of humans to technological development.

**Summary of trust in AI research.** This study synthesizes representative research on TAI (see S1 Table), identifying benevolence, reliability, functionality, transparency, accessibility, and emotional support as key factors in trust formation. However, several gaps remain in the existing literature.

First, studies on the multidimensional interactions of trust are lacking. Most research isolates individual types of trust without considering the interplay between them. Second, the role allocation and coexistence of humans and AI in HAC have been overlooked, limiting practical applications. Additionally, few studies address TAI in the context of university-enterprise cooperation and knowledge innovation management, neglecting the fundamental premise that human dominance in collaboration is essential for effective coexistence.

To address these gaps, this study constructs a cognitively operationalized quantitative model to systematically analyze how different types of TAI influence various cognitive levels. Specifically, the study aims to:

a. **Unveil the trust-cognition matching mechanism**: Investigate how different types of TAI allocate tasks between lower-order cognition (remembering, understanding, applying) and higher-order cognition (analyzing, evaluating, creating).

b. **Identify key influencing factors**: Determine which TAI types play a dominant role in shaping different cognitive levels within HAC.

c. **Analyze interactive effects**: Hypothesize that different TAI types exhibit interactive effects and propose a human-AI role allocation strategy based on complementarity and inhibition.

### Conceptual framework

Based on the above literature analysis, this study proposes a conceptual model. integrating the four Level 2 dimensions of TAI (HLT, FT, CT, ET) and their different effects on cognitive outcomes as defined by Bloom's taxonomy. See Fig 1, where the grey circles represent the significant interaction term factors.

### Methodology

#### Establishing the impact of trust in AI on cognition

In this study, TAI is categorized into four secondary dimensions: HLT, FT, CT, and ET. These dimensions further comprise six sub-dimensions and a total of 14 indicator items, resulting in a high-dimensional model. The multidimensionality and complexity of TAI are embedded throughout HAC and exert significant influence on it [7,11], indicating the presence of multiple collinearities.

Gradient Descent (GD) is one of the most essential optimization algorithms in machine learning, widely used for training various models. Its core objective is to minimize the loss function, thereby optimizing model parameters to improve prediction accuracy. One significant application of GD is in regression operations. Compared to traditional regression methods, GD is particularly effective in handling multicollinearity and demonstrates high stability when dealing with high-dimensional models. Moreover, GD is fundamentally an optimization algorithm designed to minimize loss functions rather than directly

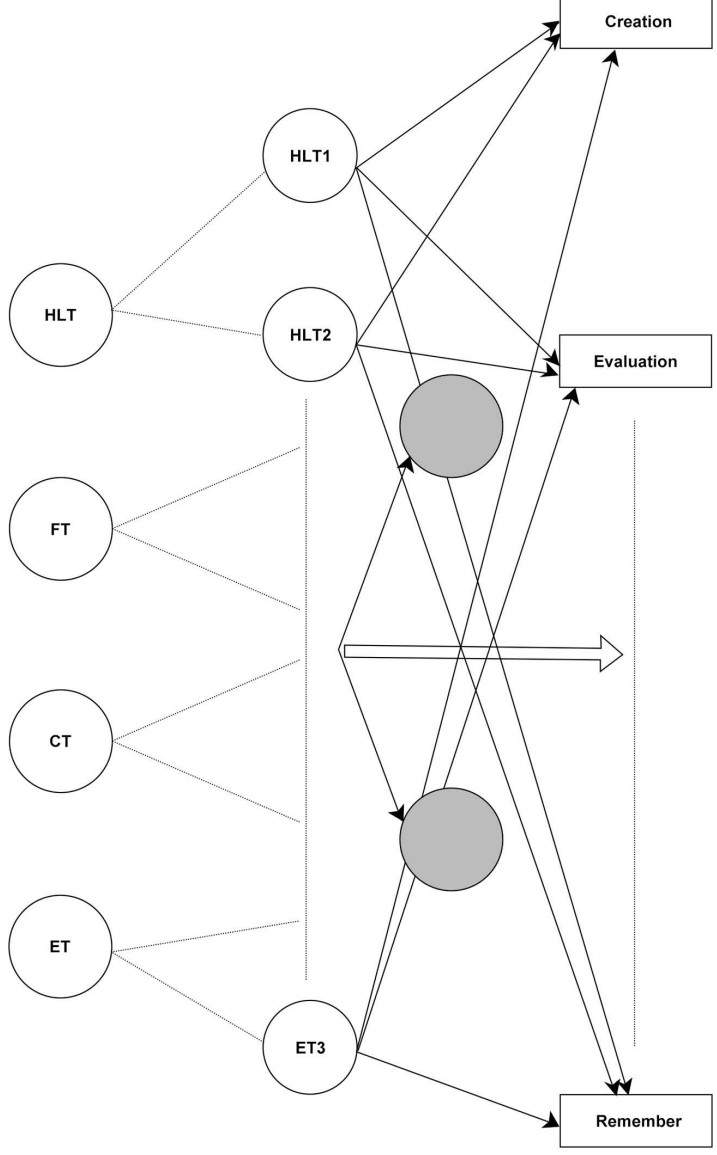

**Fig 1. Trust in AI and the modelling of cognitive taxonomy.**

explain causal relationships. The relationship between trust and cognition is not a simple causal link, as highlighted by Okamura and Yamada [9], who found that both excessive and insufficient trust impact human-AI relationships. Lastly, the relationship between TAI and cognition is nonlinear, with certain types of trust potentially exhibiting threshold effects on cognition. Therefore, this study employs Gradient Descent to model the influence of collinear TAI dimensions on cognition.

However, GD also has certain limitations, including sensitivity to initial values, the risk of overfitting, and requirements for large-scale data. To mitigate these limitations, this study implements different combinations of iteration counts and learning rates, conducting multiple experiments for fine-tuning. Overfitting experiments are then performed on the optimized parameters to prevent overfitting issues. Finally, the confirmed tuning parameters undergo multiple tests (100~400 iterations) to address sensitivity to initial values. These measures enhance the robustness of the machine learning process [40,41].

TAI exhibits multidimensionality and complexity. The specific indicators form a second-order construct, including reliability, benevolence, functionality, transparency, accessibility, and emotional connection. Based on these indicators, this study establishes the concept of TAI, with $X$ representing the feature matrix of TAI. The feature matrix is denoted as $X \in R^{n \times p}$, where $n$ is the sample size and $p$ is the number of features.

In the context of HAC, AI profoundly impacts cognition in collaboration and learning. The importance of different levels of cognition for innovation and collaboration varies significantly across different application scenarios. Therefore, we designate $Y$ as the target matrix, where $Y \in R^{n \times m}$ represents the target matrix ($n$ is the sample size, $m$ is the output type).

In gradient descent, the learning rate and the number of iterations is set. In this study, the learning rate $\propto$ is set between 0.1 and 0.0001, primarily to test the stability of the training process rather than the training time. The number of iterations is chosen to be between 500 and 5000, based on empirical rules, with the training stopping when improvement in the validation set loss is observed [42]. Therefore, within the set number of iterations, the error value $E$ is calculated (Eq. (1)), where $\hat{Y}$ is the predicted value, $Y$ is the actual value, and $\theta$ is the regression coefficient matrix.

$$E = \hat{Y} - Y = X\theta - Y \tag{1}$$

Establishing gradient regression between $X$ and $Y$ Eq. (2), we derive the partial derivatives with respect to the regression coefficients $\theta$, where $T$ represents the matrix transpose (transpose):

$$Gradients = \tfrac{1}{n} X^T E\theta = \theta - \alpha Gradients \tag{2}$$

Based on the entire sample size, a loss function Eq. (3) is established to comprehensively evaluate the predictive performance of the overall data.

$$MSE = \tfrac{1}{2n} \sum_{i=1}^{n} E_j^2 \tag{3}$$

Finally, we will conduct further analysis using the GD method to examine interactive effects. This analysis aims to address the gap in "multidimensional interaction research," as existing AI-related TAM studies primarily focus on individuals' acceptance of AI while overlooking the synergistic or conflicting effects between different types of trust. Specifically, we will determine whether these trust types enhance or offset each other. The research method involves deriving coefficients for different trust types across various cognitive levels using Equations (1–3). By ranking the coefficients and analyzing their signs, we can identify interaction patterns among trust types. We will select the top-ranked indicators with the largest absolute coefficients that exhibit opposite relationships. If a trust type's main effect is positive while its interaction term coefficient is negative, it indicates a conflict between the two. Conversely, if the interaction term is positive, it suggests synergy, meaning both trust types jointly enhance cognitive performance. In Eq. (4), $X_i$ represents a single independent variable and $X_j \times X_k$ an interaction term.

$$y = \beta_0 + \sum_i \beta_i X_i + \sum_j \beta_j \cdot (X_j \times X_k) + \varepsilon \tag{4}$$

By employing the GD method, we can gain a deeper understanding of the impact of TAI on cognition and learning behaviors. Furthermore, interaction analysis facilitates a more comprehensive understanding of role allocation within HAC.

## Sample and data collection

The primary subjects of this study are business and management interns, mainly from Wuhan Technology and Business University. This research and its related surveys are strictly regulated under the university-level project (Project No. XJ2023000501) and are subject to approval and oversight by the university's research department. Data collection took

place from 2022 to 2023, covering three major financial industrial parks in Wuhan: Xiaoguishan Financial Industrial Park, Huangjinkou Industrial Park, and the East Lake High-Tech Development Zone. The selected interns specialize in finance and business administration, fields that emphasize critical professional skills such as data analysis and forecasting, as well as strong collaboration abilities. The cultivation and acquisition of these skills significantly impact the labor market during its current transformation [14,43]. Moreover, the AI applications associated with these skills—such as quantitative analysis and risk assessment—are highly representative in HAC (HAC) scenarios. Additionally, these interns participate in cognitive internships and relevant coursework between their sophomore and junior years, making them an ideal representative sample for this study. Refer to Table 1 for detailed data.

## Measurement of trust in AI

This study primarily assesses TAI through a questionnaire survey. There is no universally accepted TAI scale due to the complexity of its dimensions and the influence of diverse application scenarios. Based on previous research, this study integrates and refines existing scales, with specific details provided in Table 2.

To ensure the reliability and validity of the questionnaire, statistical tests were conducted using SPSS. The results indicate a Cronbach's alpha of 0.88 and a KMO value of 0.926. Furthermore, key statistical indices such as KMO, AVE, and CR all meet the required standards (see S2 Table), demonstrating that the questionnaire design and refinements are both stable and effective.

## Quantification of cognitive taxonomy

For the quantification of cognitive taxonomy, this study employs the After-Action Review (AAR) method, following the approach of Keise and Arthur [44], which is particularly suitable for corporate training and management. Detailed design specifications of this method are provided in S3 Table. To assess the stability and validity of the data, statistical tests were conducted using SPSS. The results indicate a KMO value of 0.79, a total variance explained rate of 51% (with the number of factors set to 1), an AVE of 0.51, and a CR value of 0.86. These indicators demonstrate that the quantified cognitive data exhibit good convergent validity and possess effective predictive capability.

## Data analysis

### MSE test

This study employed MATLAB to test five combinations of learning rates and iteration counts (400 runs for each combination). The resulting average Total Mean Squared Error (TMSE) is shown in Table 3. According to the test results, the smallest TMSE was observed when the learning rate was set to 0.0001 and the number of iterations was 5000. The other

Table 1. Distribution of demographic characteristics of respondents (N = 408).

| Statistical variable | Form | Percentage (%) |
|---|---|---|
| Genders | male | 30 |
| | female | 70 |
| Age | 18~29 | 95 |
| | other | 5 |
| Educational background | undergraduate | 84 |
| | other | 16 |
| Experience in the use of AI | yes | 98 |
| | no | 2 |

**Table 2. Questionnaire structure and item composition.**

| Primary Concept | Definition | Secondary Dimension | Measurement Items (Likert 1–6) | Factor | Source |
|---|---|---|---|---|---|
| Human-like Trust | Reliance on AI's anthropomorphic features | Benevolence | HLT1: "I believe this AI system is designed to help users, not out of selfish interest." | 0.82 | Mayer et al., 1995; Choung et al., 2022 |
| | | | HLT2: "Smart (AI) technology cares about our well-being." | 0.87 | |
| | | Reliability | HLT3: "Smart (AI) technology keeps its promises and fulfils its commitments." | 0.87 | |
| | | | HLT4: "Smart (AI) technology is honest and does not mis-use the information and advantages it has over its users." | 0.85 | |
| Functionality Trust | Belief in AI's ability to perform tasks effectively | Competence & Comprehensiveness | FT1: "Smart (AI) technology works well to do what I ask it to do effectively." | 0.92 | Choung et al., 2022 |
| | | | FT2: "Smart (AI) technology has the functionality needed to fulfil critical tasks." | 0.9 | |
| | | | FT3: "Smart (AI) technology is competent in its area of expertise." | 0.88 | |
| Cognitive Trust | Rational evaluation of AI's logic and design | Transparency | CT1: "I can understand how AI makes decisions in work tasks." | 0.89 | Jian et al., 2000 |
| | | | CT2: "I can clearly see how AI technology works when it interacts with me." | 0.88 | |
| | | Tangibility | CT3: "Smart (AI) technology's interface design is appealing." | 0.91 | Chi et al., 2021; Choung et al., 2022 |
| | | | CT4: "I would find it easy to let Smart (AI) technology do what I want it to do." | 0.05 | |
| Emotional Trust | Emotional bonding and perceived empathy | Human-AI Emotional Interaction | ET1: "I think the AI behaves like a human when interacting with me." | 0.88 | Chi et al., 2021 |
| | | | ET2: "I think the mental effort required to interact with AI is reasonable." | 0.91 | |
| | | | ET3: "I have an emotional attachment to AI social service bots." | 0.85 | |

Note: This paper uses a 6-point Likert scale: 1 Strongly Disagree, 2 Comparatively Disagree, 3 Disagree, 4 Agree, 5 Comparatively Agree, 6 Strongly Agree. The aim is to reduce neutral views.

**Table 3. Total Mean Squared Error (Mean TMSE) for Five Parameter Combinations (400 Runs Each).**

| | Research Setting Values | test 1 | test 2 | test 3 | test 4 | test 5 |
|---|---|---|---|---|---|---|
| **Learning rate** | 0.01 | 0.1 | 0.01 | 0.001 | 0.01 | 0.0001 |
| **Iterations** | 1000 | 100 | 500 | 3000 | 2000 | 5000 |
| **Mean TMSE** | 3.1060 | 3.1158 | 3.0749 | 3.0359 | 3.0946 | 2.9619 |

Note: The measurements are from Matlab.

four combinations yielded similar results, with a Mean TMSE around 3 and relatively small differences. Subsequently, an overfitting test was conducted using the learning rate of 0.0001 and 5000 iterations. The training and validation sets were split in a ratio of 8:2, and the number of experimental runs was set to 400. Although the training error steadily decreased throughout the training process, the validation error (MSE) began to increase in the later stages, which is a likely indication of overfitting (see Fig 2). Therefore, the parameter setting of a learning rate of 0.0001 and 5000 iterations was excluded from further analysis.

By comparing the results of four experiments, this study set the learning rate to 0.01 and the number of iterations to 1000. The Mean TMSE for the test set was 3.106. The loss curves for the six cognitive levels are shown in Fig 3, which indicate that the MSE values for the analysis and application levels are relatively low.

## Hyperparameter tuning

To evaluate the stability of the model across parameter tuning (ranging from 0.01 to 1000), we conducted 100 experimental runs. This approach aims to minimize the influence of random parameter initialization and enhance the robustness of the machine learning model (see Fig 4, S4 Table). Based on the analysis results, we found significant differences in the effects of different independent variables (types of TAI) on various cognitive levels. Overall, FT plays a positive role in cognitive transformation within HAC, particularly contributing to higher-order cognition. HLTalso shows a certain degree of positive influence on lower-order cognition. Notably, the trust variables with the most pronounced negative impact are mainly concentrated in the category of CT, especially CT1 and CT2, which have significant negative effects on multiple cognitive levels. ET primarily contributes positively to higher-order cognition.

## Interactive impact analysis

According to Fig 4, we selected the indicators with the largest (positive and negative) contribution rates in each level of the cognitive taxonomy to verify the conflict and compensation effects in interactive relationships. Using the gradient descent

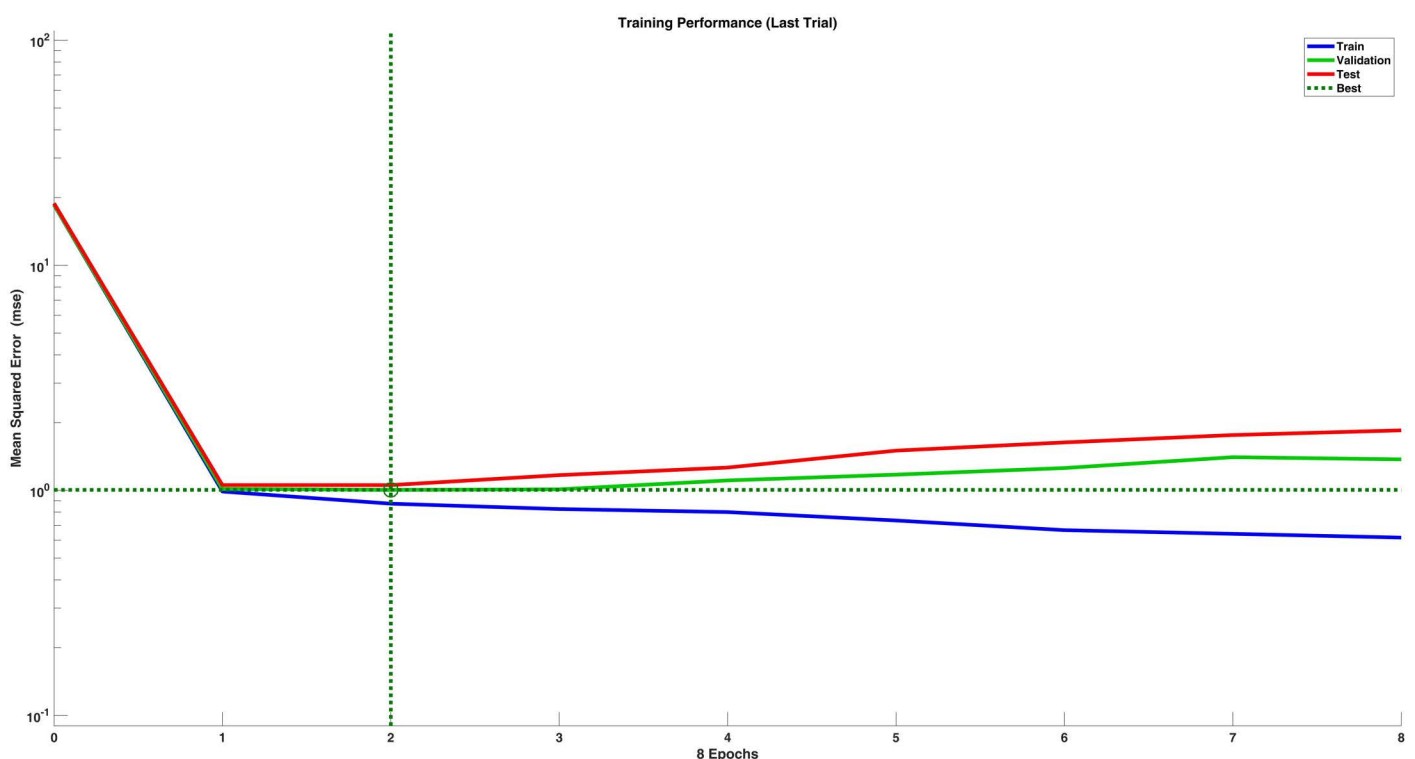

**Fig 2. Overfitting detection with a learning rate of 0.0001 and an iteration number of 5000 (400 runs).**

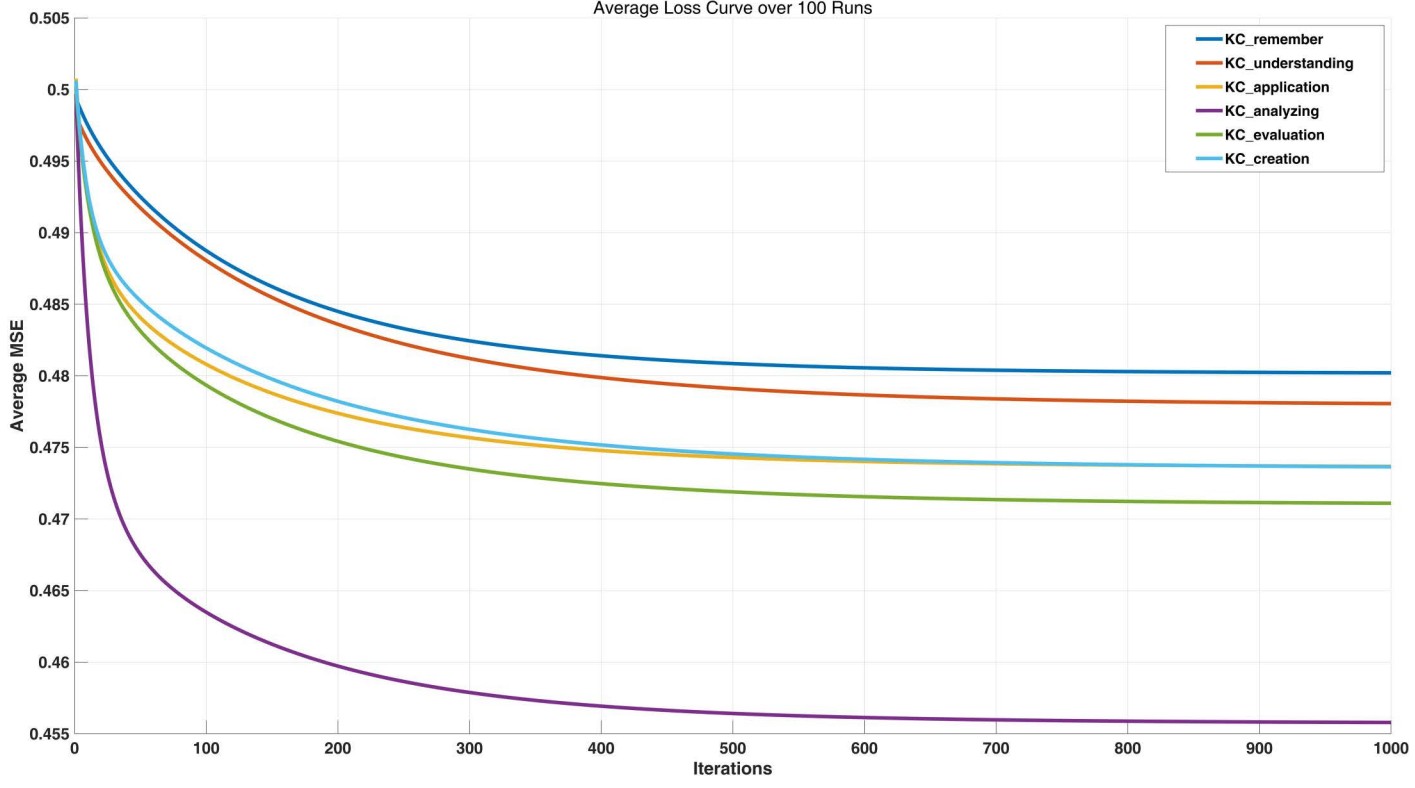

**Fig 3. Rate of change of loss function (0.01, 1000, 100 runs).**

method, we set the learning rate to 0.001 and the number of iterations to 2000. To address the sensitivity to initial values, we conducted 100 test iterations.

From the results of the interactive effect analysis (see Table 4), all interaction effects were negative. FT (competence and comprehensiveness) and ET played a crucial role in higher-order cognition, while CT (transparency) and FT (competence and comprehensiveness) exhibited an inhibitory effect.

## Discussion

### Summary of key findings

Building on prior research [7,38], this study reveals several trends and comparative insights regarding HAC, particularly the mechanisms promoting higher-order cognition and the inhibitory effect of AI transparency on cognitive performance.

First, we consolidate key findings showing that the facilitation of higher-order cognition is best supported through the synergistic interaction of emotional support and functional reliability in AI. Positive and significant effects were observed from benevolent trust (HLT1 and HLT2), FT, and ET in enabling advanced cognitive activities during collaboration. Notably, HLT and FT enhanced individual creativity, consistent with the findings of Ritala et al. [45]. Furthermore, both FT and ET strongly contributed to individuals' evaluative and analytical capacities. These results not only align with Chi et al. [38] but also underscore the role of affective interaction and system reliability as foundational mechanisms for AI to facilitate complex cognitive tasks.

While previous literature emphasizes transparency as a key attribute in fostering TAI, our findings suggest a paradox: transparency may impede cognition, especially for users operating at lower cognitive levels. This supports critiques raised

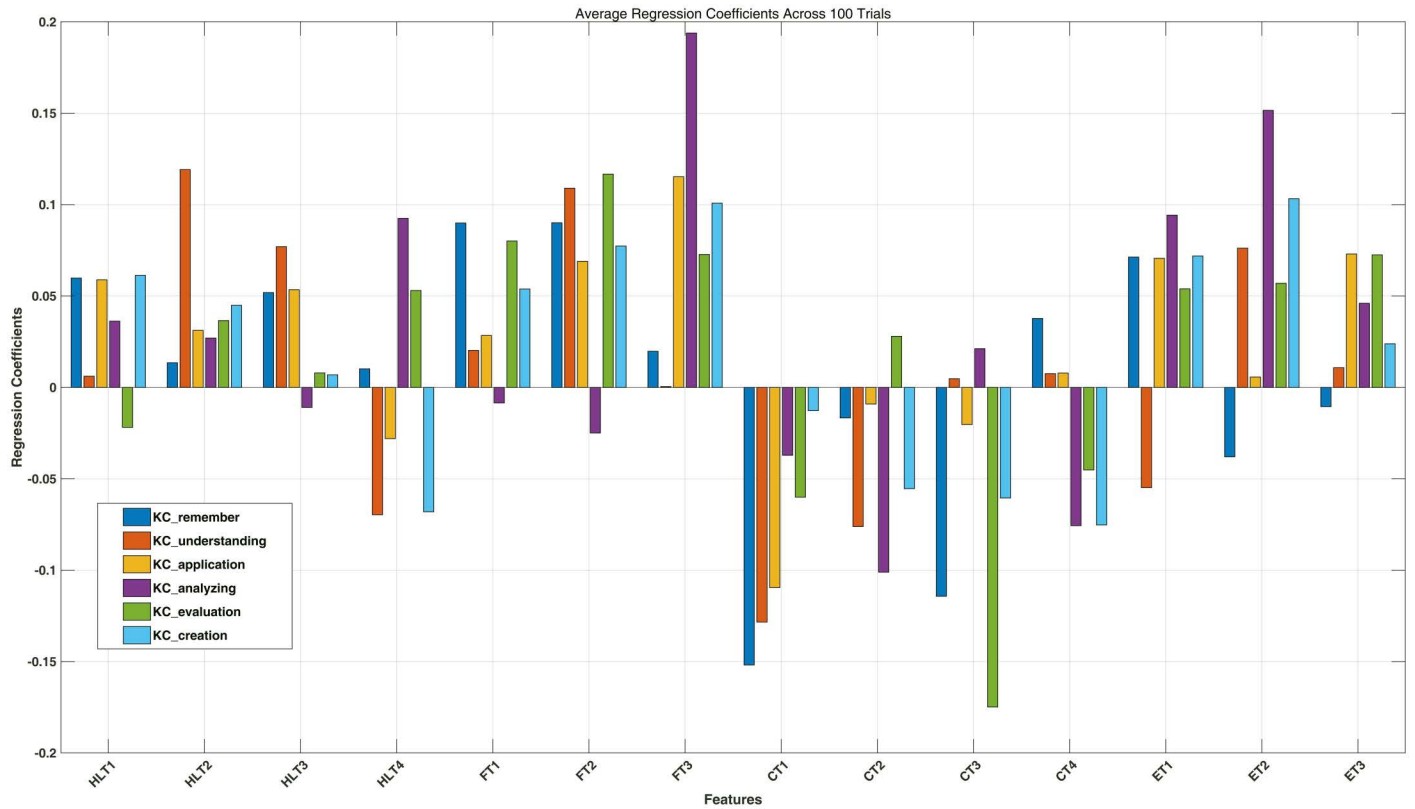

**Fig 4. Distribution of regression coefficients for 100 experiments of parameter tuning.**

by Riley et al. [37], Luyten [46], and Clemente-Suárez et al. [25], who warn of information overload and overreliance on AI systems, which can undermine self-efficacy and cognitive engagement. Our study provides empirical support for this paradox by focusing on role allocation in HAC, challenging the prevailing optimism in many human–AI interaction models regarding transparency.

Interaction effect analysis further reveals that FT and ET jointly enhance higher-order cognition, while an inhibitory interaction exists between transparency (CT) and FT. This highlights the intricate interplay among trust dimensions in AI. On the one hand, Georganta et al. [11] argue that high-quality information alleviates user anxiety, which resonates with our findings. On the other hand, our results challenge the linear assumption that greater transparency is always better, echoing Jia et al. [18], who emphasized that excessive transparency—through information overload or diffusion of responsibility—can suppress creativity in individuals with low self-efficacy. Thus, this study suggests a need to reconceptualize TAI dimensions through the lens of human–AI role distribution, recognizing that their effects are not simply additive but often exhibit compensatory or conflicting dynamics in collaborative contexts.

## Methodological innovation and implications

Departing from conventional linear or hypothesis-driven approaches [7,38], this study employs machine learning to model complex, nonlinear interaction patterns. To ensure robustness, we conducted extensive parameter tuning and repeated trials, thereby mitigating the risk of results being driven by stochastic initialization. This methodological design offers a more reliable means of capturing nonlinearities and compensatory effects.

**Table 4. Analysis of interactive impact coefficients.**

| Model | Variable | Mean_Coefficient |
|---|---|---|
| remember | Intercept | 0.060629 |
| | CT1 | 0.095882 |
| | FT1 | 0.088403 |
| | FT2 | 0.095753 |
| | CT1 x FT1 | −0.016696 |
| | CT1 x FT2 | −0.018129 |
| understanding | Intercept | 0.066647 |
| | HLT2 | 0.099611 |
| | FT2 | 0.1104 |
| | CT1 | 0.093336 |
| | HLT2×FT2 | −0.0081357 |
| | CT1×HLT2 | −0.012741 |
| | CT1×FT2 | −0.01674 |
| application | Intercept | 0.07351 |
| | FT3 | 0.12988 |
| | CT1 | 0.10979 |
| | FT3 x CT1 | −0.026638 |
| analyzing | Intercept | 0.04987 |
| | FT3 | 0.096906 |
| | ET2 | 0.086978 |
| | FT3 x ET2 | −0.011763 |
| evaluation | Intercept | 0.072916 |
| | CT3 | 0.099263 |
| | FT2 | 0.14388 |
| | CT3×FT2 | −0.027555 |
| creation | Intercept | 0.052705 |
| | FT3 | 0.10102 |
| | ET2 | 0,088735 |
| | FT3×ET2 | −0.015698 |

Moreover, the discovery of both compensatory and inhibitory interactions among different types of TAI suggests that trust mechanisms within the Technology Acceptance Model (TAM) are not merely additive. This nuance is frequently overlooked in TAM-based research, and our findings contribute to a more comprehensive understanding of trust as a dynamic and multidimensional construct in HAC.

# Conclusion

## Theoretical contributions

This study provides a fresh perspective on the construction mechanism of TAI in HAC from a cognitive standpoint, empirically exploring the role distribution within the human-AI co-existence relationship. Specifically, in the process of AI-empowered knowledge conversion, we utilized different types of TAI to explain the differential impact of AI on human cognition. Unlike previous studies [17,18], this research further uncovers the dual effects of TAI, such as the inhibitory role of transparency (CT) on cognition and the positive interaction effects between FT and ET. These findings were empirically

validated, filling gaps in both human-AI co-existence research and the study of AI-empowered knowledge collaboration and innovation.

Secondly, inspired by prior studies [3,4,9], this research further explores the interactive nature of TAI. A review of earlier research shows that studies on the impact of AI on human behavior or cognition are either lacking empirical support or fail to address the multidimensional interactions. This paper addresses these gaps by further investigating the synergistic and inhibitory factors of TAI in HAC.

Finally, by introducing machine learning to model the multidimensional interactions within the HAC relationship, this study makes a bold attempt to integrate multidimensional interaction algorithms into the Technology Acceptance Model. Through parameter tuning and robustness enhancement, we ensure stable output of research results, which provides a valuable reference for future studies.

## Practical contributions

The conclusions drawn from the data analysis suggest that AI has a dual effect on cognitive levels within HAC. Based on this, we argue that optimizing TAI can facilitate better role allocation within HAC. Specifically, role distribution within HAC can be approached from a cognitive level perspective. FT2, ET1, and ET2 all have a positive impact on higher-order cognition, while transparency has an inhibitory effect across all cognitive levels. This model provides a basis for improvement in intelligent education, knowledge collaboration innovation, and the development of human-AI co-existence environments.

The inhibitory effect of transparency on cognition has inspired us to consider that a healthy HAC should be complementary, rather than dependent. Transparency can create a sense of dependence in lower-skilled users and lead to blindness and conformity among employees with low self-determination. Information overload can reduce decision-making capabilities [18]. As Luyten [46] noted, digitization has deteriorated human reading abilities. However, effective training and the cultivation of self-management capabilities can be effective solutions to these issues.

## Limitations and future research

This study has some limitations. First, the construction of knowledge networks is the result of HAC [10]. There may be interaction effects between human trust and TAI, which require further exploration. The focus of this study is primarily on measuring human attitudes toward AI, which is why the regression coefficients are relatively low. Clearly, the cognitive level achieved through HAC is not solely attributed to AI. Future research should include human trust and even human physiological, psychological, and behavioral indicators to reduce noise.

Second, in studying the impact of TAI on innovation, this study's sample size did not account for the contextual differences across industries. The sample was mainly drawn from the financial and related sectors, yet AI technologies exhibit diverse applications across different industries [i.e., 8]. As a result, the findings may lack generalizability for cross-industry guidance. Moreover, TAI types may vary in their weight across different contexts [47]. Therefore, future research should consider cross-industry comparisons.

Third, there is subjectivity in the quantification of cognition in this study. Although the data quantification comes from different advisors and corporate evaluations, with relevant judgment criteria, it cannot be entirely guaranteed that the data are free from subjectivity. In future quantitative studies, AI text mining techniques [48] should be employed, with evaluations conducted through a fully third-party assessment approach.

Fourth, this study lacks a focus on the dynamic nature of AI technologies. The data collection period (2022–2023) coincided with the early rise of GAI technologies. Future research should further explore the potential impact of GAI and other emerging technologies on HAC and TAI dimensions while validating the generalizability of these findings across broader industry and cultural contexts.

Furthermore, one of the conclusions of this study is that transparency in TAI has an inhibitory effect on cognition. However, this conclusion needs to be further explored. GenAI technology has alternative properties to cognition that may have a debilitating effect on higher-order cognition. Future in-depth discussion of this conclusion needs to be conducted with the help of multi-task comparisons.

Finally, the study's respondents primarily consisted of recent university graduates and early-career professionals, whose limited work experience may not fully capture the patterns of AI's cognitive impact in specific industries. Future research should conduct comparative studies across individuals with varying levels of work experience to better understand how AI influences cognition across different professional backgrounds.

## Supporting information

**S1 Table. Summary of Trust in AI research literature.**
(DOC)

**S2 Table. Reliability, validity, and convergence analysis of Trust in AI questionnaire items.**
(DOC)

**S3 Table. Quantitative cognitive.**
(DOC)

**S4 Table. Regression results for 100 experiments with parameter tuning (learning rate: 0.01, iterations: 1000).**
(DOC)

**S5 File. Fundamental dataset of the study.**
(XLS)

**S6 File. Operable MATLAB Code.**
(DOC)

## Author contributions

**Conceptualization:** Weizheng Jiang , Chun Liu.

**Data curation:** Dongqin Li.

**Formal analysis:** Weizheng Jiang .

**Funding acquisition:** Dongqin Li.

**Methodology:** Weizheng Jiang .

**Project administration:** Chun Liu.

**Resources:** Dongqin Li.

**Software:** Chun Liu.

**Supervision:** Chun Liu.

**Writing – original draft:** Weizheng Jiang .

**Writing – review & editing:** Dongqin Li, Chun Liu.

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
