## [Decision Letter · Decision Letter 0]

Dear Dr. Weizheng,

Thank you for submitting your manuscript to PLOS ONE. After careful consideration, we feel that it has merit but does not fully meet PLOS ONE’s publication criteria as it currently stands. Therefore, we invite you to submit a revised version of the manuscript that addresses the points raised during the review process.

I appreciate the authors for considering PLOS ONE as the venue for their work. My decision regarding the revision is contingent upon (1) obtaining retrospective ethical approval to ensure compliance with ethical standards, (2) providing bilingual (including English) language in the supplementary data for broader accessibility, and (3) addressing the expert reviewers' comments in a detailed, point-by-point manner. Some major concerns are, but not limited to, a "sketchy" literature review, research gap articulation, hard-to-follow writing structure, theoretical framework clarification, the usefulness of the gradient descent method, data collection periods, participants (limited to financial interns), and the connection and relevance of your study to the current body of literature (discussion). 

We look forward to receiving your revised manuscript.

Kind regards,

Simon Dang

Academic Editor

PLOS ONE

Journal Requirements:

2. You indicated that ethical approval was not necessary for your study. We understand that the framework for ethical oversight requirements for studies of this type may differ depending on the setting and we would appreciate some further clarification regarding your research. Could you please provide further details on why your study is exempt from the need for approval and confirmation from your institutional review board or research ethics committee (e.g., in the form of a letter or email correspondence) that ethics review was not necessary for this study? Please include a copy of the correspondence as an ""Other"" file.

Additional Editor Comments: none

Reviewers' comments:

Reviewer's Responses to Questions

**Comments to the Author**

1. Is the manuscript technically sound, and do the data support the conclusions?

Reviewer #1: Yes

Reviewer #2: Partly

Reviewer #3: Yes

2. Has the statistical analysis been performed appropriately and rigorously?

Reviewer #1: Yes

Reviewer #2: No

Reviewer #3: Yes

3. Have the authors made all data underlying the findings in their manuscript fully available?

Reviewer #1: Yes

Reviewer #2: Yes

Reviewer #3: Yes

4. Is the manuscript presented in an intelligible fashion and written in standard English?

Reviewer #1: Yes

Reviewer #2: No

Reviewer #3: Yes

Reviewer #1: This research delves into the influence of TiAI on HAC by employing the SECI model and Bloom’s cognitive taxonomy. It enriches the trust theory through highlighting the role of TiAI in knowledge collaboration and spillover. This study holds certain theoretical and practical significance�contributing to the existing body of knowledge in the relevant academic field.

Comments:

1.The paper fails to disclose the relevant experiences of the subjects in the application or study of artificial intelligence. The knowledge background of the subjects in the AI domain has the potential to exert a notable influence on the final research conclusions.

2.The data collection period of the questionnaire spanned from 2022 to 2023. However, during 2024 and 2025, there have been remarkable breakthroughs in the field of artificial intelligence, particularly in GAI. The general public's perception of AI - based tools has also undergone substantial changes. Thus, it is necessary to question whether this study still retains its timeliness and relevance?

3.The subjects of this study are data sourced from financial interns. It is necessary to clarify the reasons for choosing financial interns rather than students from other disciplines. Moreover, it is essential to elucidate whether there are any close correlations between the research conclusions and the financial domain. Additionally, inquiries should be made regarding whether the final research conclusions are applicable to other fields.

Reviewer #2: Hi Author(s),

Thank you for submitting your work to PLOS ONE. The paper offers some insights for the field of human-AI literature. Please find my feedback below, which I hope will help you enhance your work.

1. Abstract

The abstract lacks clarity and a comprehensive overview, making it difficult to grasp the paper’s main contributions. I recommend restructuring it to clearly outline:

1. The research problems being addressed.

2. The objectives of the study.

3. The methodology employed (e.g., data collection techniques).

4. The key findings of the research.

5. The implications for both practice and theory.

6. Suggestions for future research.

This structured approach will provide readers with a concise yet thorough understanding of the paper’s significance.

2. Introduction

The introduction does not clearly articulate the paper's motivation and would benefit from a more comprehensive literature review. It is recommended that each paragraph focus on a specific research gap so that, collectively, they build a coherent narrative demonstrating how your research addresses these gaps. The authors should also identify the research questions in the introduction part.

Additionally, the paper overlooks two seminal works in AI by Huang and Rust (2021, 2024). These studies present an AI framework that spans from mechanical intelligence to cognitive reasoning (thinking intelligence) and affective understanding (feeling intelligence). Incorporating these references could provide critical context and strengthen the discussion on AI trust.

3. Literature Review

Upon reviewing the section, I found the structure of the writing somewhat difficult to follow. It is not clear what has been accomplished in the fields of cognitive quantification, human-AI interaction and AI trust, nor is it evident how your research addresses the existing gaps from a theoretical perspective.

I recommend that the authors:

• Clarify the Theoretical Framework: Clearly identify and highlight the key theories that underpin the research. A critical synthesis of previous findings should be provided to illustrate how your work builds upon and extends existing knowledge. Kindly take a look at previously published work at Plos One by Okamura & Yamada, 2020.

• Enhance Table 1: The current presentation in Table 1 is vague. It should explicitly state the findings from the reviewed papers and detail how these findings contribute to or contrast with the contributions of your study.

• Develop a Conceptual Model: The paper would benefit significantly from the inclusion of a conceptual model and the formulation of hypotheses. This would help in showcasing the relationships between the key constructs of AI trust and in framing the research contributions more coherently.

Incorporating these suggestions will help clarify your research direction and enhance the overall coherence and impact of the paper.

3. Methodology

I can see the authors indicated the rationale behind not applying for Ethical Approval. However, every research involving human participants always requires ethical approval no matter what fields they are focusing on. The role of reviewers is to provide feedback on your work, so I would not touch that aspect; the task is left for the handling Editor.

The use of the gradient descent method in this study raises important questions regarding its necessity and appropriateness. While the paper mentions that gradient descent offers advantages like computational efficiency and handling nonlinear data, it does not provide a clear justification for why this specific method was chosen over more conventional regression techniques. Given the nature of your data and research objectives, is the complexity of gradient descent truly required? Traditional statistical methods such as linear or logistic regression, which are easier to interpret and widely accepted in similar research contexts, might suffice.

The explanation of the independent (TiAI dimensions) and dependent variables (cognitive levels) is superficial. While constructs like human-like trust, functionality trust, cognitive trust, and emotional trust are mentioned, their operational definitions, measurement scales, and specific indicators are not clearly detailed. This lack of clarity makes it difficult to assess the validity and reliability of the variables used. As indicated in the study that developed a trust scale for AI trust, I would suggest the authors take a look at EFA and CFA.

The methodology depends on self-reported questionnaire data to assess trust in AI and cognitive behaviours. This shows potential biases, including social desirability bias and self-perception inaccuracies. How could the authors reduce those biases during the data collection process? We need a very clear justification here.

4. Results and Discussion

I have no further comment regarding the results section till the issues in the methodology section are resolved.

The discussion should tie the results back to the original research objectives or gaps identified in the literature review. For example, while the study identifies different dimensions of AI trust and their impact on cognitive processes, it does not clearly articulate how these findings advance our understanding of human-AI collaboration (HAC) or contribute to existing theories.

References

Okamura, K., & Yamada, S. (2020). Adaptive trust calibration for human-AI collaboration. Plos one, 15(2), e0229132.

Huang, M. H., & Rust, R. T. (2024). The caring machine: Feeling AI for customer care. Journal of Marketing, 00222429231224748.

Huang, M. H., & Rust, R. T. (2021). Engaged to a robot? The role of AI in service. Journal of Service Research, 24(1), 30-41.

Reviewer #3: Thank you for the opportunity to review the paper titled “Understanding AI Trust Dimensions through Quantitative Cognition: Implications for Human-AI Collaboration’’.

This is an interesting study and I commend the authors on taking the road least travelled. I have some comments to improve the manuscript below.

First, I believe that the abstract is not well-structured. I think the research aims are not stated frankly and specifically, making it difficult for readers to decide to read the full article. The research method is quite vague when lacking statement about the number of samples and scope. Originality and Values are summarized from the results of the article, so it clearly indicates the implications for both academic and practical circles.

The introduction provides some foundation but could benefit from further analysis, structure, and clarity to enhance its impact. It is recommended that the text be divided into more specific paragraphs: (1) Provide an overview of the context of the study, (2) Outline the importance of the study, (3) Highlight the gap in the existing literature, (4) State the specific objectives and implications of the study. Especially the paragraph about the gaps of the article should be more carefully cared for because I do not see the attractiveness and urgency of the gaps that you are simply listing. I recommend that you add to the arguments or calls to strongly emphasize the need for the filling of each gap you mentioned.

The theoretical foundations could benefit from a more comprehensive exploration of an updated literature review to align with recent advancements in the field. Only a few adequate reviews of recent literature are used in this research. You can consider changing literature that is older than 2022, most of the documents you use are quite old (citation 13-42) and this reduces the reliability of a study in a fast-moving field like AI. Please check and add more discussions from the most recent scholars.

Demographic information should be presented in detail into the table form in the sample section.

The presentation of results in the paper aligns with the analysis performed, providing clarity in reporting findings.

The discussion section, while commendable, is seen as ambiguous and lacking depth. The discussion part is where you delve into the meaning, significance, and relevance of your findings. It should be focused on discussing and evaluating what you discovered, demonstrating how it pertains to your literature review and research objectives, and presenting an argument in favor of your ultimate conclusion. How do you interpret these findings, and how do they compare to previous studies in the field? This study should compare the novelty to current and previous research for each conclusion, and then interpret the research piece in a real-world perspective.

The conclusion section should be restructured as following: theoretical and practical implications, then limitations and future research. The paper attempts to identify implications for theory but it’s still shallow, which needs more justifications and aligns with the results.

I appreciate your dedication to this research topic, and I believe that addressing these points will contribute significantly to refining the manuscript. While I acknowledge the potential significance of your work, I recommend the major revisions of the manuscript in its current form. I am confident that your revisions will strengthen the scholarly merit of your contribution.

Thank you for your understanding and efforts in advancing this research.

Best regards,

* End of reviewer comments *

**Do you want your identity to be public for this peer review?** For information about this choice, including consent withdrawal, please see our Privacy Policy

Reviewer #1: No

Reviewer #2: No

Reviewer #3: No

---

## [Author Response · Author response to Decision Letter 1]

20 Mar 2025

Dear Editor,

We sincerely appreciate your time and effort in reviewing our manuscript and providing constructive feedback. We have carefully addressed all the concerns raised and revised our manuscript accordingly. Below is our response to the key issues mentioned in your decision letter:

1. Retrospective Ethical Approval: We have obtained institutional approval and the consent of all participants. Additionally, we have submitted the necessary documentation as proof of ethical approval.

2. Bilingual Supplementary Data: We have revised the supplementary materials to include bilingual content (including English) to enhance accessibility for a broader readership.

3. Point-by-Point Response to Reviewers: We have carefully considered the reviewers' comments and provided detailed responses to each point. Moreover, we have made significant improvements to the manuscript, including:

1) Strengthening the literature review to provide a more comprehensive background.

2) Clearly articulating the research gap and contribution.

3) Refining the writing structure for better readability.

4) Clarifying the theoretical framework.

5) Elaborating on the relevance and applicability of the gradient descent method.

6) Specifying the data collection periods and justifying the selection of financial interns as participants.

7) Enhancing the discussion to better connect our findings with the existing literature.

We appreciate the opportunity to improve our manuscript and look forward to your further evaluation. Please let us know if any additional modifications are required.

Reviewer #1: This research delves into the influence of TiAI on HAC by employing the SECI model and Bloom’s cognitive taxonomy. It enriches the trust theory through highlighting the role of TiAI in knowledge collaboration and spillover. This study holds certain theoretical and practical significance�contributing to the existing body of knowledge in the relevant academic field.

Comments:

1. The paper fails to disclose the relevant experiences of the subjects in the application or study of artificial intelligence. The knowledge background of the subjects in the AI domain has the potential to exert a notable influence on the final research conclusions.

Response: We did not explicitly present the participants’ prior experiences in the application or study of artificial intelligence in the paper. However, we did collect this information during the study. The decision to omit it was based on the following considerations:

1. This variable was not a significant predictor in our data analysis model;

2. To streamline the manuscript and enhance readability;

3. Most students had already taken relevant courses and participated in cognitive internships before their senior year, equipping them with foundational AI knowledge.

Nevertheless, we acknowledge the reviewer’s suggestion, as prior experience and knowledge structure may theoretically influence the dependent variable. To address this, we have incorporated relevant details in the "Sample and Data Collection" section to strengthen the rigor of our study.

2. The data collection period of the questionnaire spanned from 2022 to 2023. However, during 2024 and 2025, there have been remarkable breakthroughs in the field of artificial intelligence, particularly in GAI. The general public's perception of AI - based tools has also undergone substantial changes. Thus, it is necessary to question whether this study still retains its timeliness and relevance?

Response: AI applications in the domain of GAI have indeed garnered significant attention. However, the core conclusions of this study remain both universal and timely. First, deep learning and neural network algorithms have been widely integrated into various applications since as early as 2012, and the rapid advancement of modern AI has been largely driven by the synergy of General Purpose Machine Learning (GPML), Data Generation, and Domain-Specific Knowledge Structures (Taddy, 2018). Second, although AI technology continues to evolve, the theoretical framework of AI trust remains applicable, particularly in the context of HAC and its role in fostering innovation. Furthermore, while GAI technology witnessed a breakthrough in 2024, the fundamental functionalities of AI and the mechanisms of human-AI trust have remained consistent across various application scenarios.

Additionally, our research team acknowledges the rapid evolution of AI technologies. As a result, we have addressed this consideration in the outlook section of the manuscript:

“Fourth, the data collection period of this study spanned from 2022 to 2023, during which GAI technology was still in its early stages. Future research should further explore the potential impact of generative AI and other emerging technologies on human-AI collaboration and AI trust dimensions, while also validating the generalizability of these findings across a broader range of industries and cultural contexts.” (line: 565-569)

Taddy, M. (2018). The technological elements of artificial intelligence (No. c14021). National Bureau of Economic Research. DOI: 10.3386/w24301

3. The subjects of this study are data sourced from financial interns. It is necessary to clarify the reasons for choosing financial interns rather than students from other disciplines. Moreover, it is essential to elucidate whether there are any close correlations between the research conclusions and the financial domain. Additionally, inquiries should be made regarding whether the final research conclusions are applicable to other fields.

Response: First, our sample includes not only finance interns but also students majoring in business administration. (The lack of clarity in our manuscript’s wording may have led to some misunderstanding.) Students in these disciplines typically possess strong data analysis and forecasting skills, which are increasingly valuable in today’s evolving labor market (Alekseeva et al., 2021). In particular, these skills are directly linked to critical tasks such as quantitative analysis and risk assessment in the financial sector—domains that serve as highly representative and typical cases for studying Human-AI Collaboration (HAC).

Second, we emphasize the essential role of these skills in AI applications. AI-driven financial tasks, such as quantitative analysis and risk assessment, not only require advanced data processing capabilities but also demand a deep understanding of business contexts and decision-making processes. By selecting interns from these disciplines, our study effectively captures the real-world challenges and opportunities associated with HAC in professional environments.

Finally, regarding the generalizability of our findings, while this study focuses on the financial sector, its core conclusions—particularly those concerning skill complementarity and task allocation in human-AI collaboration—extend beyond finance. Other industries, including healthcare, law, and education, are also experiencing rapid AI integration and shifts in skill demands. Therefore, our findings provide valuable insights applicable to a broad range of fields.

To address this consideration, we have refined the Sample and Data Collection section of the manuscript as follows:

“The selected interns specialize in finance and business administration, fields that emphasize critical professional skills such as data analysis and forecasting, as well as strong collaboration abilities. The cultivation and acquisition of these skills significantly impact the labor market during its current transformation [14,43]. Moreover, the AI applications associated with these skills—such as quantitative analysis and risk assessment—are highly representative in HAC scenarios.” (line: 334-340)

Alekseeva L, Azar J, Giné M, Samila S, Taska B. The demand for AI skills in the labor market. Labour Economics. 2021;71: 102002. doi:10.1016/j.labeco.2021.102002

Reviewer #2: Hi Author(s),

Thank you for submitting your work to PLOS ONE. The paper offers some insights for the field of human-AI literature. Please find my feedback below, which I hope will help you enhance your work.

1. Abstract

The abstract lacks clarity and a comprehensive overview, making it difficult to grasp the paper’s main contributions. I recommend restructuring it to clearly outline:

1. The research problems being addressed.

2. The objectives of the study.

3. The methodology employed (e.g., data collection techniques).

4. The key findings of the research.

5. The implications for both practice and theory.

6. Suggestions for future research.

This structured approach will provide readers with a concise yet thorough understanding of the paper’s significance.

RESPONSE: We have further refined the summary and followed the structure you provided.

The details are as follows:

“Human-AI collaborative innovation relies on effective and clearly defined role allocation, yet empirical research in this area remains limited. To address this gap, we construct a cognitive-level AI trust framework to describe and explain its interactive mechanisms in human-AI collaboration, specifically its complementary and inhibitive effects. Specifically, we examine the alignment between AI trust and different cognitive levels, identifying key drivers that facilitate both lower-order and higher-order cognition through AI. Furthermore, by analyzing the interactive effects of multidimensional AI trust, we explore its complementary and inhibitive influences. We collected data from finance and business administration interns using surveys and the After-Action Review (AAR) method and analyzed them using the gradient descent algorithm. The findings reveal a dual effect of AI trust on cognition: while functional and emotional trust enhance higher-order cognition, the transparency dimension of cognitive trust inhibits cognitive processes. These insights provide a theoretical foundation for understanding AI trust in human-AI collaboration and offer practical guidance for university-industry partnerships and knowledge innovation.”

2. Introduction

The introduction does not clearly articulate the paper's motivation and would benefit from a more comprehensive literature review. It is recommended that each paragraph focus on a specific research gap so that, collectively, they build a coherent narrative demonstrating how your research addresses these gaps. The authors should also identify the research questions in the introduction part.

Additionally, the paper overlooks two seminal works in AI by Huang and Rust (2021, 2024). These studies present an AI framework that spans from mechanical intelligence to cognitive reasoning (thinking intelligence) and affective understanding (feeling intelligence). Incorporating these references could provide critical context and strengthen the discussion on AI trust.

Response: We appreciate this recommendation and have made substantial revisions based on the feedback from both Reviewer 2 and Reviewer 3. Specifically, we have implemented the following improvements:

Structural Adjustments: We have reorganized the introduction to establish a clearer and more coherent narrative. The revised structure now explicitly addresses the background, research significance, identified gaps, specific objectives, and broader implications of our study.

Emphasis on Research Gaps: We have strengthened the discussion on AI trust research gaps, focusing on the following key aspects:

The underlying mechanisms of human trust in AI remain insufficiently understood.

Most existing studies lack a multidimensional perspective on AI trust interactions.

There is a notable absence of empirical evidence in research on human-AI coexistence theories, particularly concerning role allocation in collaborative settings.

Justification for Cognitive Hierarchies: We have elaborated on why cognitive hierarchies serve as an effective framework for interpreting human-AI collaboration and AI trust. Additionally, we highlight the advantages of this approach in addressing the identified research gaps.

Incorporation of Seminal Works: We have carefully reviewed and integrated insights from Huang and Rust (2018, 2021, 2024), as well as Okamura and Yamada (2020). Their discussions on AI’s progression from mechanical intelligence to cognitive reasoning and affective understanding have provided valuable context for our study. In particular, their findings on threshold effects in AI trust have reinforced the need for further exploration of multidimensional AI trust interactions, particularly in the context of human-AI role allocation.

These refinements significantly enhance the clarity and rigor of our introduction, ensuring that our study is well-positioned within the broader AI trust literature.

3. Literature Review

Upon reviewing the section, I found the structure of the writing somewhat difficult to follow. It is not clear what has been accomplished in the fields of cognitive quantification, human-AI interaction and AI trust, nor is it evident how your research addresses the existing gaps from a theoretical perspective.

I recommend that the authors:

• Clarify the Theoretical Framework: Clearly identify and highlight the key theories that underpin the research. A critical synthesis of previous findings should be provided to illustrate how your work builds upon and extends existing knowledge. Kindly take a look at previously published work at Plos One by Okamura & Yamada, 2020.

• Enhance Table 1: The current presentation in Table 1 is vague. It should explicitly state the findings from the reviewed papers and detail how these findings contribute to or contrast with the contributions of your study.

• Develop a Conceptual Model: The paper would benefit significantly from the inclusion of a conceptual model and the formulation of hypotheses. This would help in showcasing the relationships between the key constructs of AI trust and in framing the research contributions more coherently.

Incorporating these suggestions will help clarify your research direction and enhance the overall coherence and impact of the paper.

Response:

The work of Okamura & Yamada (2020) has provided us with significant insights, particularly regarding the potential threshold effects of AI trust—where exceeding or falling below a certain level may either enhance or inhibit trust. However, our research focuses on the multidimensional nature of AI trust. Given this perspective, our team believes it is essential to empirically validate the interactive effects of AI trust across multiple dimensions. This multidimensional interaction study does not contradict our previous research; rather, it further clarifies and expands on the dynamics of harmonious human-AI relationships.

Regarding the summary of AI trust research in Table 1, we have moved the original Table 1 to the supplementary materials and provided a more detailed discussion of relevant AI trust studies. In fact, we initially offered an in-depth interpretation of Table 1 within the manuscript. However, we opted to streamline this section for a specific reason: PLOS ONE has a strong empirical and methodological orientation, with a writing style that emphasizes conciseness and clarity. Therefore, in the Revised Manuscript with Track Changes, we have made the following adjustments: The original Table 1. TiAI types and constructs has been relocated to S1. Table AI trust research literature collation. Additionally, we have expanded our review of recent AI trust research, particularly studies published since 2022, in the AI Trust Dimensions section.

To further clarify our research objectives and contributions to addressing existing gaps, we have restructured the Theoretical Foundations section.

Reviewer 2 suggested: "Develop a Conceptual Model: …………more coherently." Instead of constructing a predefined conceptual model, we have chosen to refine and elaborate on our research objectives. The reason for this approach is that the relationship between AI trust and cognitive hierarchies remains unclear, and we aim to avoid imposing any premature assumptions. In the Summary of AI Trust Research (lines 227–249), we have revised and articulated our research objectives as follows:

• Unveil th

---

## [Decision Letter · Decision Letter 1]

PLOS ONE

Dear Dr. Weizheng,

Thank you for submitting your manuscript to PLOS ONE. After careful consideration, we feel that it has merit but does not fully meet PLOS ONE’s publication criteria as it currently stands. Therefore, we invite you to submit a revised version of the manuscript that addresses the points raised during the review process.

Editor's comments:

I thank the authors for their meticulous revisions and for being receptive to comments from the reviewer team and myself. The work has been refined significantly. However, there are still some minor points that require your attention. Below I outline some of the points worth your additional refinement.

1. Make sure to ensure consistent citation formatting throughout (e.g., "Fig." instead of "Fig"; standardize table references) and uphold high resolution for clarity and professional presentation.

2. All measurement items in Table 2 must be fully presented to facilitate readership.

3. I concur with R2 that a conceptual model must be presented way before any empirical results to avoid the seemingly P-HARking.

4. I recommend comprehensively reassessing all the content in the appendix and supplementary files to include necessary content to support your main text so that readers don't have to jump back and forth to find them. Those in the appendix should only serve to extend and clarify some major points explained in the main text if readers wish to pursue further.

5. The discussion should highlight the uniqueness of your study. You are doing a great job of expounding on how the findings 'align' with extant studies. How about the 'contrasting' and novel findings parts? This will make your work much stronger and is well-positioned in the literature.

6. I notice many abbreviations and inconsistent terminology (e.g., AI trust). I would advise the use of the full form throughout the paper, except for the results or content in tables, which often come with explanations/unabbreviated forms.

Other than that, I think we are on the right track toward publication.

We look forward to receiving your revised manuscript.

Kind regards,

Simon Dang, Ph.D.

Academic Editor

PLOS ONE

Journal Requirements:

Reviewers' comments:

Reviewer's Responses to Questions

**Comments to the Author**

Reviewer #1: (No Response)

Reviewer #2: All comments have been addressed

Reviewer #3: All comments have been addressed

2. Is the manuscript technically sound, and do the data support the conclusions?

Reviewer #1: Yes

Reviewer #2: Yes

Reviewer #3: Yes

3. Has the statistical analysis been performed appropriately and rigorously?

Reviewer #1: Yes

Reviewer #2: Yes

Reviewer #3: Yes

4. Have the authors made all data underlying the findings in their manuscript fully available?

Reviewer #1: No

Reviewer #2: No

Reviewer #3: Yes

5. Is the manuscript presented in an intelligible fashion and written in standard English?

Reviewer #1: Yes

Reviewer #2: Yes

Reviewer #3: Yes

Reviewer #1: The author has made commendable efforts to address reviewers' concerns by enhancing discussions on sample representativeness, ethical approval statements, and methodological transparency. However, there is room for improvement in several areas:

1. The paper could benefit from a more thorough analysis of the potential impact of generative AI systems (e.g., GPT-4) on cognitive inhibition effects, including empirical validation in future research.

2. To bolster technical reproducibility, it is recommended to provide supplementary pseudo-code or actual implementation code for the gradient descent algorithm.

3.The original text alternates between "TiAI (Trust in AI)" and "AI trust," which may cause confusion. It is advised to standardize the terminology throughout the paper.

4.Low-resolution images should be replaced with high-definition versions to enhance clarity and professionalism.

5.Citation formats for figures and tables need standardization; for example, "Fig" should be consistently written as "Fig.", and table references should follow a uniform style (e.g., "Table 1" rather than "S1 Table").

Reviewer #2: Hi Author(s),

Thank you for submitting your revision to PLOS One. I appreciate the improvements you've made. For the next revision, I suggest that the author(s) highlight all changes in yellow. This will help the reviewers easily see what has been modified.

One major issue I found in this revision is the table, figure and images both in the appendix and supplementary files. They contain many formatting flaws and are difficult to see. I request the author(s) to revise this aspect carefully.

I would focus on the answer that you provided to the comments:

I can see the justification you provided for including Table 1. However, I think that Table 1 in the main body of the manuscript should highlight the key findings from the included studies. Specifically, it should address the relationships between the constructs. At present, Table 1 does not offer much insight.

The authors mentioned against making premature assumptions, so a conceptual framework is not included. I would like to see a clear justification for the chosen method in the literature review section to support this claim. In the Methodology section, the author(s) stated, "To achieve this, we construct a framework for AI trust, which consists of four major dimensions and fourteen specific items." If that is the case, why is a conceptual model not included to provide a clearer presentation?

I appreciate the authors’ explanation of the gradient method; that makes sense to me. The methodology section is well explained.

I would expect the Discussion section would include a clear theoretical contribution, practical contribution and methodological contribution. Currently, the discussion section just simply rewords the findings.

Thank you! I look forward to receiving your second revision.

Reviewer #3: All the revisions have addressed all my comments. Therefore I suggest accepting the publication of this article.

**Do you want your identity to be public for this peer review?** For information about this choice, including consent withdrawal, please see our Privacy Policy

Reviewer #1: No

Reviewer #2: No

Reviewer #3: No

---

## [Author Response · Author response to Decision Letter 2]

21 May 2025

Reviewer & Editor Comment 1

Ensure consistent citation formatting throughout (e.g., "Fig." instead of "Fig"; standardize table references) and uphold high resolution for clarity and professional presentation.

We appreciate the detailed suggestions from the editor and reviewers. We have standardized the formatting of all in-text figure and table citations (e.g., “Fig.”, “Table”), and all images have been replaced with high-resolution (300 dpi) versions to ensure clarity and professionalism. All the graphs were tested by PACE and the results are as follows:

Additionally, in response to Reviewer 1’s concern regarding the formatting and naming of “S1 Fig” and “Table” references, we have made the necessary adjustments. The former S1 Table has been renamed Supplementary Table S1, and we have standardized the naming conventions of all supplementary figures and tables to fully comply with PLOS ONE’s formatting guidelines for supporting information.

Reviewer & Editor Comment 2

All measurement items in Table 2 must be fully presented to facilitate readership.

We have now fully presented all measurement items in Table 2, including: Primary Concept, Definition, Secondary Dimension, Measurement Items, Factor Loading, and Source, to enhance transparency and facilitate reader comprehension.

Reviewer & Editor Comment 3

The conceptual model must be presented before any empirical results to avoid the appearance of P-HARKing.

We accepted this suggestion and have repositioned the conceptual model figure to the theoretical foundation section, accompanied by a new subsection titled “Conceptual Framework” (Line 252–258). This clarifies the theoretical logic and helps avoid any appearance of post-hoc hypothesis generation (P-HARKing).

Reviewer & Editor Comment 4

Reassess all content in the appendix and supplementary files. Only include content that extends or clarifies the main text.

We have made the following adjustments to the supplementary materials:

1. S1 Table has been revised to emphasize key findings and research gaps in the literature on trust in AI (TAI), corresponding to Section 2 of the manuscript. As the content serves primarily as literature background, we have kept it in the supplementary files.

2. Per Reviewer 1’s suggestion, we have uploaded the MATLAB source code for the gradient descent algorithm to enhance reproducibility. Additionally, we removed the original S4 and S5 Tables (parameter tuning results) and replaced them with a TMSE test based on 400 iterations (see Table 3). Relevant content in Lines 382–420 has been revised accordingly.

Reviewer & Editor Comment 5

The discussion should highlight the uniqueness of your study. Move beyond alignment with existing work to contrast and novelty.

We have restructured the Discussion section into two parts: “Summary of Key Findings” and “Methodological Innovation and Implications.” We emphasized not only alignment with previous studies but also novel findings, such as the inhibitory effect of transparency on lower-level cognition. Moreover, we discussed the nonlinear interaction mechanisms of trust in AI (TAI) within human-AI collaboration (HAC), offering new insights from the perspective of role allocation and cognitive stratification.

Reviewer & Editor Comment 6

Inconsistent terminology (e.g., "AI trust"). Prefer full form “trust in AI” (TAI) throughout.

We have standardized terminology usage across the manuscript by replacing “AI trust” with “trust in AI (TAI)” throughout, except in tables and figures where the abbreviation is retained for conciseness.

Reviewer 1 Comment – Generative AI

Suggest further reflection on how GenAI (e.g., GPT-4) may influence cognitive inhibition, and recommend empirical validation in future work.

Thank you for the valuable suggestion. We have added content in the conclusion section (Lines 548–552), highlighting that GenAI technologies may exhibit distinct properties influencing cognition, especially potential inhibitory effects that warrant further empirical exploration.

Reviewer 1 Comment – Provide Pseudo-code or Implementation

To enhance reproducibility, suggest uploading pseudo-code or actual implementation code for gradient descent.

We have uploaded the MATLAB implementation code as suggested, to enhance the reproducibility of the machine learning procedures used.

Reviewer 2 Comment – Clarify Table 1

Table 1 should better highlight key findings and inter-construct relationships.

Thank you! We believe this refers to S1 Table in the supplementary materials, not the main text Table 1. Accordingly, we have restructured S1 Table to better illustrate key relationships between constructs and support the theoretical foundation discussed in Section 2.

Reviewer 2 Comment – Emphasize Conceptual Framework Early

The conceptual framework should be clearly presented and justified early in the paper.

We have repositioned the conceptual model to the theoretical section and added a new subsection titled “Conceptual Framework” (Lines 252–258), to clarify its theoretical origins and its relevance to our research design.

Reviewer 2 Final Comment – Discussion Should Include Three Contributions

The discussion should clearly state theoretical, methodological, and practical contributions.

We have revised the Discussion section to explicitly present three types of contributions: theoretical, methodological, and practical. This highlights the novel directions our study introduces to research on trust in AI (TAI) and human-AI collaboration (HAC).

---

## [Decision Letter · Decision Letter 2]

Understanding dimensions of trust in AI through quantitative cognition: Implications for human-AI collaboration

PONE-D-24-58698R2

Dear Dr. Weizheng,

We’re pleased to inform you that your manuscript has been judged scientifically suitable for publication and will be formally accepted for publication once it meets all outstanding technical requirements.

Kind regards,

Simon Dang, Ph.D.

Academic Editor

PLOS ONE

Additional Editor Comments (optional):

Reviewers' comments:

Reviewer's Responses to Questions

**Comments to the Author**

Reviewer #2: All comments have been addressed

2. Is the manuscript technically sound, and do the data support the conclusions?

Reviewer #2: Yes

3. Has the statistical analysis been performed appropriately and rigorously?

Reviewer #2: Yes

4. Have the authors made all data underlying the findings in their manuscript fully available?

Reviewer #2: Yes

5. Is the manuscript presented in an intelligible fashion and written in standard English?

Reviewer #2: Yes

Reviewer #2: Hi Author(s),

Thank you for all the work that you have put into this manuscript. The author(s) addressed all of my comments.

I'm happy to support the publication of this manuscript.

Well done!

**Do you want your identity to be public for this peer review?** For information about this choice, including consent withdrawal, please see our Privacy Policy

Reviewer #2: No

---

## [Editor Report · Acceptance letter]

PONE-D-24-58698R2

PLOS ONE

Dear Dr. Weizheng,

I'm pleased to inform you that your manuscript has been deemed suitable for publication in PLOS ONE. Congratulations! Your manuscript is now being handed over to our production team.

Kind regards,

on behalf of

Dr. Simon Dang

Academic Editor

PLOS ONE